# Thimet Oligopeptidase Biochemical and Biological Significances: Past, Present, and Future Directions

**DOI:** 10.3390/biom10091229

**Published:** 2020-08-24

**Authors:** Emer S. Ferro, Mayara C. F. Gewehr, Ami Navon

**Affiliations:** 1Pharmacology Department, Biomedical Science Institute, University of São Paulo, São Paulo, SP 05508-000, Brazil; ferrari.mayaracalegaro@gmail.com; 2Department of Biological Regulation, The Weizmann Institute of Science, Rehovot 7610001, Israel; ami.navon@weizmann.ac.il

**Keywords:** peptide metabolism, peptidase, protease, proteasome, protein-protein interaction, microRNA

## Abstract

Thimet oligopeptidase (EC 3.4.24.15; EP24.15, THOP1) is a metallopeptidase ubiquitously distributed in mammalian tissues. Beyond its previously well characterized role in major histocompatibility class I (MHC-I) antigen presentation, the recent characterization of the THOP1 C57BL6/N null mice (THOP1^−/−^) phenotype suggests new key functions for THOP1 in hyperlipidic diet-induced obesity, insulin resistance and non-alcoholic liver steatosis. Distinctive levels of specific intracellular peptides (InPeps), genes and microRNAs were observed when comparing wild type C57BL6/N to THOP1^−/−^ fed either standard or hyperlipidic diets. A possible novel mechanism of action was suggested for InPeps processed by THOP1, which could be modulating protein-protein interactions and microRNA processing, thus affecting the phenotype. Together, research into the biochemical and biomedical significance of THOP1 suggests that degradation by the proteasome is a step in the processing of various proteins, not merely for ending their existence. This allows many functional peptides to be generated by proteasomal degradation in order to, for example, control mRNA translation and the formation of protein complexes.

## 1. A Brief Historical Perspective 

THOP1 was initially identified as a metalloendopeptidase (also known for a long time as endopeptidase 24.15, EP24.15), purified from the soluble fraction of rat brain homogenates with maximal catalytic activity seen at about pH 7.8 [1]. After its isolation and characterization, THOP1 was considered to be similar to the already identified peptidases named Pz-peptidase, kininase A, or endo-oligopeptidase A [2,3,4,5,6]. Substrate specificity and several biochemical properties of Pz-peptidase, kininase A or endo-oligopeptidase A were shared by THOP1, including the activity stimulated by low concentrations of reducing compounds (e.g., 0.1–0.5 mM dithiothreitol or 1–2 mM β-mercaptoethanol) but inhibited at higher concentrations [7,8,9]. THOP1 is strongly inhibited by o-phenanthroline, but its complete inhibition by ethylenediaminetetraacetic acid (EDTA) is only achieved after extensive dialyses [1]. Additionally, the metal-chelated inactive form of THOP1 can be reactivated by several divalent metal ions, including Zn^2+^, Co^2+^, and Mn^2+^ [1]. THOP1 cDNA cloning allowed for the identification of the metallopeptidases’ HEXXH motif [10,11]. Later site-directed mutagenesis studies confirmed that THOP1 shared the typical catalytic mechanism of metalloproteases [12,13]. The zinc ion coordination occurs by two histidine residues from the HEXXH motif plus an additional upstream glutamic acid residue (E501). The glutamic acid residue from the HEXXH holds a water molecule at the fourth zinc ion coordination position by a hydrogen bond. This well placed and activated water molecule makes a nucleophilic attack during the peptide bond hydrolysis of the substrates of THOP1. Molecular cloning has also shown that fifteen cysteine residues are present in the rat THOP1 protein. The THOP1 crystal structure revealed that seven out of the fifteen cysteine residues were solvent accessible, and two were found inside the catalytic site cleft; no intramolecular disulfide bonds were observed [14]. These data contributed to understanding one of the most distinguishable characteristics of THOP1, which is its so-called thiol-sensitivity. THOP1 preparations become less active under non-reducing conditions because of the formation of inactive oligomers mediated by intermolecular disulfide bridges [8]. This oligomerization can be avoided in a reductive medium (for example, presence of low concentrations of dithiothreitol or β-mercaptoethanol). By replacing the solvent-exposed cysteines with serine residues this oligomerization is impaired, which was elegantly shown by the double (C246S/C253S) or the triple (C246S/C248S/C253S) rat THOP1 mutants that were fully active in the absence of reducing agents [8]. These data suggest that low concentrations of reductive thiol compounds activate THOP1, breaking the intermolecular disulfide bonds and restoring the amounts of monomeric active THOP1 from the soluble but inactive small oligomer (dimers and trimers) pools of THOP1. The low concentrations of reductive thiol reagents is now accepted as the molecular mechanism through which THOP1 enzymatic activity is “activated” [8]. Concerning the inhibition of THOP1 by larger concentrations of thiol compounds, it is well accepted that this is due to the known extreme avidity of the SH group present on these reagents with the zinc ions of the THOP1 catalytic activity site. THOP1 immunoprecipitated from cells submitted to oxidative challenge (H_2_O_2_ treatment), showed increased trimeric forms and decreased S-glutathionylated forms compared to THOP1 immunoprecipitated from control cells [9]. THOP1 maximal enzymatic activity was found to be maintained by partial S-glutathionylation. Maximal THOP1 activity was observed when THOP1’s total reduced cysteine residues were 10 ± 1, whereas minimal activity was observed when the reduced cysteine residues decreased to 7 ± 0.5; intermediate activity was founded when 14 ± 0.5 cysteine residues were S-glutathionylated. This latter mechanism suggests how THOP1 oligomerization and maximal activity is regulated in vivo [9,15].

In 1992, the International Union of Biochemistry and Molecular Biology (IUBMB) officially recommended that metalloendopeptidase/endopeptidase 24.15, pz-peptidase and endo-oligopeptidase A be jointly designated as “thimet oligopeptidase”, an acronym for the characteristic “thiol-sensitive metallopeptidase” [3,4,16]. Therefore, several publications in the period from ~1983–1992, particularly about THOP1 physiological functions on neuropeptide metabolism, certainly cause some confusion to readers that are not aware that different names were coined for the same enzyme [6,17,18,19,20,21,22,23]. It must be noted, however, that the name endo-oligopeptidase A still remains in use, associated to a yet uncharacterized enzymatic function of the structurally distinctive coiled-coil nuclear distribution element-like protein “NUDEL-oligopeptidase” [24,25,26,27,28,29,30,31].

## 2. THOP1 Substrates: Only Size Matters 

The substrate-size specificity presented by THOP1 is perhaps the most robust characteristic of this enzyme. THOP1, at the time called “brain kininase A”, was well known to hydrolyze the FS peptide bond in bradykinin (RPPGF↑SPFR) [32,33,34]. The substrate-size restriction of THOP1 was first documented showing that pretreatment of denatured kininogen with “brain kininase A” (THOP1) did not reduce the amount of trypsin-releasable bradykinin. These data suggested that the bradykinin sequence, when part of a large protein, was not a substrate for THOP1 [35]. The crystal structure of THOP1 elegantly confirmed these earlier biochemical suggestions, by describing the location of the enzyme’s active site at the base of a narrow deep channel, which limits the size of substrates that can enter to be hydrolyzed by THOP1 (Figure 1) [14].

Optimal peptide sizes cleaved by THOP1 are those containing 9–12 amino acids, however, the enzyme can cleave peptides containing from 5–22 amino acids [36,37,38]. The amino acid sequence of a potential THOP1 substrate is unpredictable, as the sub-site specificity of THOP1 is unusual toward specific amino acid residues present at certain P/P‘ positions [37,39,40]. THOP1 can cleave with similar kinetic efficiency in between amino acid residues of distinctive chemical characteristics (i.e., LR, RR, FS or PF) and also in between two distinctive residues of similar substrates (GGFL↑R↑RVQ; GGFL↑R↑RVE; GGFI↑R↑RVQ; PVNF↑K↑F↑LSH; VVYPW↑T↑Q↑RY; LVVYP↑W↑T↑Q↑RY cleaved bonds are indicated by arrow) [13,40]. Analysis of THOP1 substrate specificity using a peptide library [41] corroborates both biochemical [13,37,40] and structural [42] observations, showing that no single amino acid plays significant role in defining the substrates of THOP1 (Figure 2).

Together, these features make any oligopeptide containing 5–22 amino acids, including those optimally containing from 9–12 amino acids, possible substrates for THOP1 [37,38,40,44]. 

THOP1 interacts with most substrates with relatively high affinity (*K*_M_ varying from 0.2–10 µM), whereas the hydrolysis occurs at relatively low turnover ratios (*k_cat_* varying from 6–60 min^−1^) [37]. Neprilysin (EC 3.4.24.11; NEP), which is a typical neuropeptide degrading ectoenzyme present in brain synapses [45], has higher *K_M_* (30–300 µM) for most substrates, while the *k_cat_* varies from 1000–10,000 min^−1^ [46]. In the brain, THOP1 is mainly an intracellular enzyme located inside the nucleus (75%) and in the cytoplasm (25%) [47,48,49]. The amount of THOP1 found in the nucleus was inversely correlated with that found in the cytoplasm, suggesting that the enzyme could be mobilized from one compartment to the other [49]. Within the cytoplasm, THOP1 immunoreactivity was detected throughout perikarya and dendrites, as well as within axons and axon terminals. These results strongly suggested that THOP1 could play a major role in the hydrolysis of intracellular (intranuclear and cytoplasmic) substrates [49]. These kinetic differences together with a predominant intracellular localization, suggest that the major biological significance of THOP1 may not be related to brain neuropeptide metabolism [49].

## 3. THOP1 Provides a Major Contribution to Major Histocompatibility Complex of Class I (MHC-I) Antigen Presentation

The characterization of cytosolic peptides containing 8–9 amino acids as antigens presented by the MHC-I [50,51,52], was followed by evidence of cytoplasmic proteasomal processing of proteins to produce MHC-I peptides [53]. The seminal work associating THOP1 to MHC-I antigens was conducted in antigen-presenting cells (APCs) endogenously producing mycobacterial hsp65 antigen (J774-hsp65 cells) obtained by infecting the macrophage-like BALB/c cell line J774 with a retroviral shuttle vector [pZIPNeoSV(X)] carrying the mycobacterial gene [54]. Increasing THOP1 activity by 20% in these cells induced a significant increment in CD8 but not CD4 mediated hsp65 antigen presentation. Conversely, THOP1 inhibition produced an inhibition of CD8 mediated hsp65 antigen presentation, without affecting CD4 antigen presentation [55]. Inhibition of the proteasome, as expected, caused a significant decrease in CD8 antigen presentation, which was not reverted by overexpressing or inhibiting THOP1. These data suggest that THOP1 was acting on MHC-I antigens downstream of the proteasome [55]. An apparent paradox then developed in the field, when it was shown that THOP1 is competitively inhibited (most *Kiapp* from 0.21.5 µM) by most of the MHC-I antigens evaluated [44]. Further experiments were conducted in H-2Kb-expressing COS7 cells (COS-Kb) that presented the MHC-I antigen SIINFEKL, KVVRFDKL or ASNENMETM showing that 15–16 fold of THOP1 overexpression inhibited CD8 antigen presentation [56]. Conversely, siRNA inhibition of THOP1 expression was able to increase CD8 antigen presentation in transfected COS-Kb cells [56]. Thus, it was suggested that THOP1 inhibits the presentation of multiple antigenic peptides generated by proteasomes from full-length proteins [56]. These latter mechanisms of action of THOP1 inhibiting MHC-I antigen presentation were corroborated by Kim et al. [57]. These apparently controversial results were recently brought to a conciliatory mechanism [58]. The role of THOP1 in the production of the PRAME (PRA_190–198_), an immunodominant CTL epitope from Epstein-Barr virus protein EBNA3C and a clinically important epitope of the melanoma protein MART-1 epitopes, was in accordance with an independent description that THOP1 can both destroy and generate peptides of the HLA class I ligand lengths [38]. Thus, the role of THOP1 in MHC-I antigen processing seems to be twofold. THOP1 has a destructive role [56,57], limiting presentation of epitopes whose C termini have already been made by, for example, the proteasome, such as SIINFEKL [56]. On the other hand, THOP1 can also generate C-termini fit for binding by HLA class I via trimming of epitope precursors with an extension of the C terminus. Thus, THOP1 can either inhibit or enlarge MHC-I antigen processing depending on the antigen [58] (Figure 3). Interestingly, THOP1 has been shown to regulate the expression levels of MHC-I in primary-derived human aortic endothelial cells (HAECs), pointing to a role for THOP1 in the flow-mediated regulation of endothelial immunogenicity [59].

Recently, the generation of THOP1 C57BL/6 null mice (THOP1^−/−^) allowed possible key physiological functions of THOP1 to be investigated [60,61]. Using an experimental autoimmune encephalomyelitis (EAE), the animal model of multiple sclerosis [62], THOP1^−/−^ mice was characterized to present worse clinical behavior scores compared to WT mice, from 17th–26th day postimmunization with MOG35-55 peptide. EAE symptoms of THOP1^−/−^ were associated with increased tumoral necrosis factor alpha (TNF-α) levels in the dorsal hippocampus and spinal cord at the 26th day pos-immunization. During EAE, MHC I-restricted myelin basic protein is presented by oligodendrocytes and cross-presented by Tip-dendritic cells (DCs) to CD4^+^ T cells, which then spread it to myelin-specific CD8^+^ T cells that directly recognize oligodendrocytes [63]. Therefore, these data may suggest that the phenotype of THOP1^−/−^ mice in EAE is due to increased antigen presentation, leading to increased inflammation and TNF release. In the future, novel studies could suggest THOP1 as a target for the treatment of specific neurodegeneration and autoimmunity diseases. Together, these studies opened a possibility that THOP1 was important for post-proteasomal metabolism of peptides.

## 4. Seminal Experiments Evidencing that Intracellular Peptides (InPeps) Generated by Proteasome Activity, were Naturally Present in Cells and Tissues

Evolutionary analysis strongly suggests that generation of InPeps by proteasome must be an ancestral function compared to the so well demonstrated proteasome antigen processing [64,65,66]. During evolution, active proteasome appears in certain bacteria and archaea, while MHC-I appears later on cartilaginous fish [66,67,68,69]. Moreover, only one from hundreds of peptides generated by proteasome activity during protein degradation, is presented as antigen by MHC-I [70,71] (Figure 3). To investigate the possible broader function of proteasome activity-generated peptides [71,72], THOP1 catalytic activity was inactivated by site-directed mutagenesis, and its recombinant inactive structured form was used in a substrate-capture assay [13]. The peptides identified by electron spray mass spectrometry (ESI-MS/MS) after using this “substrate-capture assay”, indeed were all fragments of intracellular proteins [13]. All of the first thirteen novel peptides identified using the substrate-capture assay were fragments of intracellular proteins. Chemical synthesis allowed the investigation of the kinetic constants of three hemoglobin fragments, PVNFKFLSH (hemopressin; Ki*app* 27.76 µM), VVYPWTQRY (Ki*app* 10.02 µM) and LVVYPWTQRY (Ki*app* 2.56 µM) [13]. Compared to bradykinin (100%), the relative hydrolysis ratio of PVNFKFLSH (140%), VVYPWTQRY (10.62%) and LVVYPWTQRY (0.72%) suggested similarities to what was previously observed for MHC-I antigens [44]; PVNFKFLSH was a good substrate of THOP1, while LVVYPWTQRY was a competitive inhibitor [13]. Hemopressin, when administrated intravenously, was shown to have hypotensive activity [13,73]. Later, hemopressin became the first of a large family of peptides with pharmacological inverse agonist activity over type 1 cannabinoid receptors [74,75]. Therefore, the substrate capture assay using inactive THOP1 allowed identification of a novel class of peptides, which were generally coined InPeps. Moreover, the “substrate capture assay” was critical to suggest that InPeps as proteasome activity products further processed by THOP1, were functional beyond MHC-I antigens [71]. 

With the rapid advance of nano liquid chromatography in tandem with ESI-MS/MS (LC-ESI/MS/MS), peptidomics can start to be used to identify multiple peptides in biological samples, without the need to digest the proteins and characterize the resulting peptides [76,77,78]. InPeps were further sequenced directly from the low molecular weight fraction of cells and/or tissues homogenates using liquid-chromatography in tandem with mass spectrometry (LC-MS/MS), without the need of the substrate capture assay [38,79,80,81,82,83,84]. To date, multiple research groups have identified InPeps in different human cell lines such as HEK293, MCF7 and SH-SY5Y [85], human tissues [86,87], rodents [76,77,78,88,89,90], zebrafish [91], yeast [81], and plants [82,83], even without using the “substrate capture assay” [38,79,90]. Altogether, these data suggest that peptides generated from proteasome activity must have broader function beyond MHC-I antigens [71,92,93].

The first peptidome analyses of the THOP1-transfected and mock-transfected human embryonic kidney 293 (HEK293) cells, using isotopic labeled semi-quantitative LC–ESI/MS/MS, identified 38 peptides [38]. Of these 38 peptides, 5 were decreased more than 70% and another 12 were partially decreased; these peptides were determined as being in vivo substrates of THOP1. The levels of three peptides were elevated by the overexpression of THOP1, two of these by 3-fold or more; these elevated peptides represent in vivo products of THOP1. Eighteen of the identified peptides were not altered by the overexpression of THOP1, suggesting that some HEK293 peptides are neither substrates nor products of THOP1 [38]. An opposite approach was also performed in HEK293 cells, using specific THOP1 siRNA (THOP1-siRNA) sequences that caused, approximately, a 75% reduction on the THOP1 enzymatic activity and displayed more than 95% of immunoreactivity measured by Western blot [94]. Semi-quantitative LC-ESI/MS/MS peptidome analyses were also performed on these cells, confirming that only some peptides from HEK293 cells were affected by the absence of THOP1 [94]. Importantly, all peptides identified on HEK293 peptidome analysis were derived from intracellular proteins, with subcellular localization of the precursor proteins as follows: 75% cytosolic, 12% mitochondrial and 20% nuclear [85]. These data corroborate previous biochemical data that THOP1 only hydrolyzes a specific unpredictable set of peptides, even when these peptides are in the correct range of THOP1 substrate size specificity. Moreover, these data also suggested that THOP1 intracellular substrates and products could be playing key functions within cells. 

The majority of HEK293 and and human neuroblastoma SH-SY5Y cell peptides identified during peptidome studies were likely to be generated by proteasomes, as they are fragments of intracellular proteins, mostly containing from 5 to 17 amino acids in length and having either a hydrophobic (75%), basic (12%), or acidic (8%) amino acids at their C terminus [95]. Thus, InPeps’ profile was shown to be strongly modulated by proteasome inhibition, both in HEK293 and SH-SY5Y cells and murine brain [72,80,81,96,97,98,99,100]. When considering the length of the endogenous substrates of THOP1, the majority of the substrates were peptides that were 9 to 11 amino acids long. The peptide products of THOP1 using high or low concentrations of the enzyme range from 6 to 13 amino acids, with products that averaged 10 to 11 amino acids in length [38].

Several InPeps were chemically synthesized and tested with the recombinant active purified THOP1, to further confirm that peptides altered by overexpression of THOP1 in HEK293 cells were substrates or products of the enzyme. The peptide AVAIKAMAK (AVA-9) was cleaved by purified THOP1 with the highest efficiency of all peptides studied; this peptide showed a very large decrease upon overexpression of THOP1 in HEK293 cells. The synthetic peptide AIRDIDLKNR (AIR-10) was found cleaved in vitro into AIRDIDL and KNR. Similarly, the peptide AIRDIDLKNR was found in HEK293 cell extracts to be a substrate of THOP1, and the peptide AIRDIDL was found to be a product when the cell peptide extract was incubated with the high concentration of THOP1. The peptide SAMTEEAAVAIKAMAK was found cleaved at the predicted site, between the K↑A, located four residues from the C terminus. The synthetic peptide VFDVELLKLE was also cleaved four amino acids from the C terminus at the L↑L bond, as predicted from the analysis of the substrate/product pairs in HEK293 identified peptides [38]. The peptide ELFADKVPKTAENFR was predicted to be cleaved at two sites as follows: one part being four residues from the end at the A↑E bond, and the other part being two residues from the end at the N↓F bond. The in vitro cleavage of these peptides by the recombinant THOP1, performed under zero order, was completely prevented by the THOP1 inhibitor CFP. Altogether, these results are consistent with the following model: proteasome degrades intracellular (cytosolic, nuclear and mitochondrial) proteins generating peptides containing an average of 5 to 17 amino acids and, some of these peptides can be further processed by THOP1 generating smaller peptides [38,39,40,41,42,43,44,45,46,47,48,49,50,51,52,53,54,55,56,57,58,59,60,61,62,63,64,65,66,67,68,69,70,71,72,73,74,75,76,77,78,79,80,81,82,83,84,85,86,87,88,89,90,91,92,93,94]. 

## 5. Identification of Functional InPeps Suggests that Protein Degradation by Proteasome Is Not the End, But Rather a Next Step in Protein Function

InPeps isolated using the catalytically inactive THOP1 substrate-capture assay, were shown to contain post-translational modification motifs, mostly for protein kinase phosphorylation [93]. Phosphorylation of the substrate VVYPWTQRY (VVYPW-PThr-QRY) increased the Ki ratio by 3.3-fold, while phosphorylation of a similar substrate, LVVYPWTQRY (LVVYPW-PThr-QRY), or the substrate LNETGDEPFQYKN (LNE-PThr-GDEPFQYKN), reduced the Ki by 0.38-fold [93]. Peptide LVVYPWTQRY, and to a lesser extent VVYPWTQRY, competitively inhibited phosphorylation of a protein kinase C standard substrate [92]. These results suggest that inside the cells phosphorylation of InPeps could significantly change their processing kinetics by THOP1. Moreover, these results also suggested that InPeps could be affecting signal transduction by competing with protein targets of protein kinases, and by this means be natural modulators of protein interactions within cells [71,92,101].

InPeps can be chemically synthesized to bond to a cell-penetrating peptide (cpp) through a permanent or reversible disulfide bond, in order to further investigate their possible intracellular function in several cell lines [102,103,104,105]. Reintroduced into either HEK293 or Chinese hamster ovary (*CHO-S*) cells, InPeps were shown to potentiate signal transduction of G-protein coupled receptor (GPCR) agonists such as isoproterenol and angiotensin II [106]. Some of the peptides potentiating GPCR activity were shown to bind specific proteins, suggesting a possible mechanism for InPep potentiation of signal transduction through modulation of protein-protein interactions [106]. The peptide WELVVLGKL (Pep5) from G_1_/S-specific cyclin D2 showed a 2-fold increase during the S phase of HeLa cell cycle. Pep5 (25–100 μm) induced cell death in several tumor cells (i.e., HeLa, MDA-MB-231, MCF-7, glioma C6, SKRB, SK-MEL 28, MEL 85, SBCl-2, TPC-1, Nthy-ori 3-1, and KTC-2) only when it was fused to a cell-penetrating peptide (Pep5-cpp), suggesting its intracellular function [103,105]. 

THOP1 overexpression in both HEK293 and CHO-S cells was sufficient to reduce luciferase activation triggered by isoproterenol or angiotensin II [106]. Conversely, THOP1 inhibition by siRNA potentiated beta-adrenergic signal transduction in HEK293 cells. Together, these data further corroborate the suggestion that THOP1 is involved in processing functional InPeps [38,94]. Simulating mechanical or hemodynamic forces associated with blood flow (cyclic strain) on vascular bovine arterial endothelial cells (BAECs) up-regulates THOP1 mRNA expression and enzymatic activity, in a force- and time-dependent manner [107]. Pharmacological agents and dominant-negative G protein mutants used to selectively disrupt Gi_α2/3_ greatly attenuated strain-dependent up-regulation of THOP1 in BAECs. Activation of Gi protein subunits and increased GTPase activity occur immediately (within seconds) after mechanical loading, suggesting that THOP1 could be facilitating signal transduction of mechanical stimuli in vascular endothelial cells [107]. THOP1 mRNA expression and enzymatic activity were also up-regulated in human and bovine aortic endothelial cells exposed to physiological levels of laminar shear (0–10 dynes/cm², 24–48 h) [59]. The up-regulation of THOP1 was completely prevented by pre-treatment of endothelial cells with either N-acetylcysteine, superoxide dismutase, or catalase, confirming the involvement of reactive oxygen species. Taken altogether, these findings suggest that pathophysiogical events that up- or down-regulate THOP1 enzymatic activity should change the InPep profile, which could interfere with G-protein receptor signal transduction as well as reactive oxygen species production. The novelty that InPeps generated by proteasome are functional beyond MHC-I antigen presentation, has potential to became a novel paradigm in cell biology. 

## 6. The Biological Significance of THOP1 Represented through both Neuropeptides and InPeps

THOP1 biological significance is related to its enzymatic activity, that is destroying or generating functional peptides both inside and outside of the cells. These characteristics suggest that THOP1 biological significance should be seen through its regulation of the levels of peptides, substrates or/and products. The initial isolation of neuropeptides from brain tissue, associated with selective substrate-size specificity, suggest that neuropeptides were natural THOP1 substrates [1,108,109]. Several evidences have suggested the secretion of THOP1 through an alternative constitutive and regulated secretory pathways [19,49,110,111,112,113,114,115] (Figure 4). Approximately 10% of the total cellular content of THOP1 from AtT20 or glioma C6 cells were released to the extracellular milieu, where the enzyme could be metabolizing neuropeptides either into a soluble [19,111,112] or calcium-mediated membrane-bound form [113,114,115] (Figure 4). Utilizing the mouse hypothalamic neuronal GT1-7 cell line, it was demonstrated that THOP1 exists within lipid rafts in the plasma membrane, and that the enzyme is localized to the exofacial leaflet of lipid rafts [116] (Figure 4). The non-classical secretion inhibitor glyburide, a blocker of ATP-sensitive K^+^ channels, decreased the amount of constitutively released THOP1 in cell media of GT1-7 cells, suggesting that THOP1 association with lipid rafts on the extracellular surface precedes constitutive release of the peptidase into the extracellular milieu for its action on neuropeptides [116]. THOP1 (as endo-oligopeptidase A-like, formerly EC 3.4.22.19) was reported secreting from glioma C6 cells (Ferro et al. 1993) and also from median eminence neurons [117,118]. An intact secretory pathway flow is critical for THOP1 secretion, which can be stimulated by calcium ionophore A23187 and corticotrophin-releasing hormone and blocked by brefeldin A and nocodazole [111]. These data were corroborated by confocal and electron microscopic immunogold labeling showing that in all rat brain neuronal compartments, THOP1 is mainly associated with membranes of neurosecretory elements, including Golgi cisternae, tubulovesicular organelles, synaptic vesicles, and endosomes, always facing the cytoplasmic layer of these membranes [49,110]. THOP1 physically interacts with 14-3-3 epsilon, which is an isoform of a family of ubiquitous phosphoserine/threonine scaffold proteins that organizes cell signaling and is involved in exocytosis [112]. The interaction between THOP1 and 14-3-3 epsilon occurs at the THOP1 region containing serine 644, and can be increased by THOP1 phosphorylation by PKA [112]. This interaction to 14-3-3 epsilon is believed to allow THOP1 to indirectly associate with membranes. Overexpression of 14-3-3 epsilon in HEK293 cells almost doubled the secretion of THOP1 stimulated by A23187. Treatment of HEK293 cells with forskolin, an activator of PKA, increases THOP1 co-localization with 14-3-3 epsilon and has a synergistic effect on the A23187-stimulated secretion of THOP1 that can be totally blocked by the PKA inhibitor KT5720 [112]. The final step of THOP1 secretion is tightly regulated by phospholipid bilayer asymmetry [119,120], as suggested by the reversible inhibitory effects of manganese chloride (MnCl_2_) on THOP1 A23187-stimulated secretion [112]. Altogether, these data suggest that THOP1 can be secreted and membrane associated through a constitutive and regulated unconventional secretory pathway to metabolize neuropeptides outside the cells. Recently, *The Human Reference Protein Interactome Mapping Project* described THOP1 interacting with TNF receptor-associated factor 2 (TRAF2) and Epsin-2 (EPN2) [121]. The biological implications of these interactions for THOP1 secretion or intracellular function remain uninvestigated.

Chemical inhibition of THOP1 produced changes in reproductive mechanisms [118,124,125], nociception [36,115,126] and blood pressure [127]. These are physiological functions mediated, at least in part, by neuropeptides within the optimal substrate-size specificity of THOP1, such as gonadotrophin-releasing hormone (GnRH), dynorphin A1-8, octapeptide, and bradykinin [16]. THOP1 null mice (THOP1^−/−^) allowed further elucidation of THOP1 physiological relevance to phenotypes associated with the metabolism of opioid peptides, GnRH, bradykinin and additional peptides [61]. THOP1^−/−^ mice showed higher sensitivity to the intraplantar administration of bradykinin, and under these conditions their peripheral nociceptive response in the hot plate test was much faster than that of C57BL6/N wild type (WT) mice [61]. These observations indicate that THOP1 is a rate-limiting peptidase for bradykinin inactivation in the periphery [61]. These data corroborated previous suggestions using chemical inhibition that THOP1 plays a key function inactivating bradykinin in the circulation [127], and could also be a pharmacological target for the treatment of certain types of peripherical pain. 

However, another recent possibility has been described. RNA sequencing of half a million single cells allowed for the creation of a detailed census of cell types in the mouse nervous system [128]. THOP1′s highest expression was observed at specific non-peptidergic neurons located at the dorsal root ganglia (DRG) (http://mousebrain.org/) [128]. The cell body of a pseudounipolar neuron is located within a DRG. The axon leaves the cell body (and out of the DRG) into the dorsal root, where it splits into two branches. The central branch goes to the dorsal columns of the spinal cord, where it forms synapses with other neurons. The peripheral branch travels through the distal dorsal root into the spinal nerve all the way until the skin, joints, and muscles [129]. Therefore, intracellular THOP1 could be altering bradykinin signal transduction similarly to what it does for the beta-adrenergic agonist isoproterenol [94,106]. Other genes that characteristically appeared enhanced in these same DRG neurons were natriuretic peptide type B (Nppb), transmembrane protein 233 (Tmem233), Interleukin-31 receptor subunit alpha (Il31ra), adenosine deaminase (Ada), and insulin enhancer protein ISL-2 (Isl2). Future experiments should address the function of THOP1 on controlling bradykinin signal transduction as well as the expression of these specific groups of genes in these specific neuronal cells.

WT and THOP1^−/−^ mice present similar behavior in the hot plate test commonly used to evaluate the central nervous system action of opioid drugs [130]. A higher NEP mRNA expression level was observed in the striatum of THOP1^−/−^, suggesting that the lack of THOP1 expression triggered a homeostatic compensation inducing the expression of additional peptidases for opioid peptide metabolism in specific areas of the central nervous system (CNS) [61]. These results are in agreement with the fact that peptidases are not substrate specific [131], corroborating previous suggestions by chemical inhibitors that THOP1 is important for opioid metabolism in the CNS [17,126,132]. A physiological function of THOP1 in the regulation of gonadotrophin-releasing hormone (GnRH) effects was suggested using chemical inhibitors [108,118,125,133,134,135]. Rat THOP1 phosphorylation at serine 644 by protein kinase A (PKA), dramatically changed its GnRH substrate-specificity with a seven-fold increase in both *Km* and *k*_cat_, further suggesting the physiological relevance of THOP1 for GnRH metabolism [135]. In vivo THOP1 presence was shown in the median eminence and, more importantly, secreted in the region where GnRH axon terminals and the hypophysial portal vessels are present [118]. Infusion of a THOP1 inhibitor augmented the steroid-induced LH surge in ovariectomized rats, suggesting that THOP1 can play a role in the regulation of GnRH release [118]. Intracerebroventricular administration of *N*-[1-(*R*,*S*)-carboxy-3-phenylpropyl]-AAY-p-aminobenzoate (CFP), a specific inhibitor of THOP1, resulted in increased gonadotropin secretion and increased recovery of intracerebroventricular-administered GnRH [108,133]. THOP1^−/−^ mice were viable, possessed an estrus cycle, were fertile, and had a number of puppies per litter similar to WT mice. The lack of changes in the estrous cycle together with a normal reproductive function and litters of normal numbers, are strong evidences that both GnRH and LH levels were within a regular physiological range in THOP1^−/−^ [61]. After being secreted, GnRH can be degraded in the hypothalamus and the anterior pituitary gland by two endopeptidases, THOP1 and prolyl oligopeptidase, acting in a stepwise manner [125,136]. The mRNA expression of prolyl oligopeptidase was not altered in different regions of the THOP1^−/−^ mice brain, suggesting that additional enzymes and/or additional physiological mechanisms could be involved in regulating GnRH function and/or metabolism in the absence of THOP1. Two enzymes which possessed upregulated expressions in THOP1^−/−^ were ACE1 and NEP [61]. Altogether, these results suggest that THOP1 has key functions in GnRH metabolism, which should at least affect reproduction and estrus cycle, and may be compensated by other enzymes similar to what seems to be occurring for opioid peptide metabolism.

THOP1^−/−^ mice displayed depression-like and attention deficit and memory retention phenotypes, which is known to involve neurotransmitters and neuropeptides [137]. One observation made was that THOP1^−/−^ have lower turnover ratios of 5-hydroxyindoleacetic acid 5 (5HIAA)/serotonin (5HT) in the prefrontal cortex, and of homovanillic acid (HVA)/dopamine (DA) and 4-dihydroxyphenylacetic acid (DOPAC) + HVA/DA in the striatum. It is well documented that serotonin and dopamine play important functions in neuronal communication and in many psychiatric disorders, including depression and schizophrenia, that can affect attention and cognition [137]. Moreover, in the striatum and hippocampus of THOP1^−/−^ the mRNA levels for dopamine DRD2 receptors were lower compared to WT, while high mRNA levels of serotonin 5HT2a receptor expression in the hippocampus were observed for THOP1^−/−^ compared to WT. Together, these results suggest that THOP1^−/−^ affects the mRNA expression of DA and 5HT receptors, which may induce a rebalance of DA and 5HT neurotransmitter turnover ratios in these animals. The molecular mechanisms of such phenotypes remain largely unknown, while it may be mediated by the altered metabolism of neuropeptides and/or InPeps due to the lack of THOP1.

## 7. THOP1 Plays Key Unanticipated Physiological Functions on Energy Metabolism Regulation

Human THOP1 was identified as having a role in early diabetic retinopathy [138], and genome-wide association studies identified THOP1 among the novel genetic features of cholesterol and lipoprotein metabolism [139,140]. Moreover, in a primate model of maternal obesity induced in baboons prior to pregnancy, THOP1 was identified as one of the five differentially expressed proteasome pathway genes targeted by four differentially expressed microRNAs [141]. Neurolysin (Nln), which is a closely-related THOP1 oligopeptidase [37], was recently shown to be a key enzyme for energy metabolism improving glucose uptake and insulin sensitivity in mice [142]. Together, these previous reports suggest a possible involvement of THOP1 and Nln within metabolic disorders. The physiological function of THOP1 on energy metabolism regulation was further investigated using a murine model of hyperlipidic (HD) diet-induced obesity (DIO) and insulin resistance. Mice were fed, *ad libitum*, either a standard diet (SD) or an HD, for 24 consecutive weeks. By the end of these 24 weeks, THOP1^−/−^ male mice fed an HD only gained 25% of the body weight gained by the WT also fed an HD (Figure 5). THOP1^−/−^ fed HD showed neither insulin resistance nor non-alcoholic fatty liver steatosis when compared to WT mice fed HD; these phenotypes were also seen in THOP1^−/−^ females, although, with modest differences [60]. THOP1^−/−^ mice were healthy, viable and have normal external appearance, estrous cycle, and fertility. Their litters had a normal number of animals, which cannot be visually distinguished from WT littermates [60,61]. Behavior analyses showed no alterations on locomotor activity with either THOP1^−/−^ or WT animals, which spend similar time in different zones of the open field apparatus with similar rearing behavior/amount of time crossing the central square/instances of grooming [61]. The elevated plus maze task suggested no signs of altered anxiety behavior of THOP1^−/−^ compared to WT mice [61]. In Barnes maze and novel object recognition tasks also no alterations of THOP1^−/−^ in learning/cognitive performance were observed comparing to WT [61]. Blood parameters including hemoglobin concentration, white blood cells, red blood cells, red cell distribution width, mean corpuscular volume and mean corpuscular hemoglobin concentration were similar comparing THOP1^−/−^ to WT mice [61]. Altogether, these data strongly suggest that THOP1^−/−^ mice were perfectly healthy. Thus, specific mechanisms are taking place to prevent THOP1^−/−^ mice fed with HD from weight gain. 

The reduced body weight and adipose tissue of the THOP1^−/−^ animals could be due to their distinctive fat metabolism, which did not induce higher heat production, nor was it related to higher locomotor activity across the 24 h period; these latter parameters were even lower while considering the interval from 5–10 p.m. [60]. The food, water, and caloric intakes among these different groups were similar. In addition, there were no observed differences in the total fecal lipid content of both WT and THOP1^−/−^ mice that were fed either an SD or a HD during the 24-week experiment [60]. At the highest intensity of the treadmill test, for both females and males, the THOP1^−/−^ animals ran longer than the WT animals before exhaustion [60]. At a low intensity in the treadmill tests no differences were seen when comparing THOP1^−/−^ and WT mice [60]. The higher lipolytic ability of THOP1^−/−^ mice could have contributed to a better performance on the high-intensity treadmill test, thus increasing its oxidative metabolism [60]. Among the possible signaling pathways involved in this differential phenotype was the adipose tissue adrenergic-stimulated lipolysis pathway. THOP1^−/−^ adipose tissue had increased lipolysis compared to WT, which could be related to the higher expression of β1, β2 and β3 adrenergic receptors in THOP1^−/−^ [60]. These data corroborate previous observations that altered expression of specific InPeps and THOP1 plays a role on beta-adrenergic signaling transduction pathway [94,106,107]. Altogether, these data suggest that an alteration in the InPeps profile of the adipose tissue caused by the lack of THOP1 expression, improved lipolysis, increasing its oxidative metabolism. That seems to be critical for THOP1^−/−^ mice to became resistant to DIO as well as consequentially to having neither insulin-resistance nor non-alcoholic liver steatosis [60]. 

Searching for a mechanism of action that could connect THOP1 enzymatic activity to the altered phenotype of THOP1^−/−^ mice, differences were found in InPeps profiles from the inguinal adipose tissue comparing to WT mice, either fed SD or HD [60]. There are several possibilities that InPeps could be affecting energy metabolism, one of these could be by modulating protein-protein interactions. Most of the InPep precursor proteins identified in the inguinal adipose tissue of THOP1^−/−^ mice were previously shown to be functional in energy metabolism [60]. For example, the histone-derived InPeps identified have different levels in THOP1^−/−^ and have putative post-translational lysine modification sites in their structure (AQGGVLPNIQAVLLPK and KQVHPDTGISSKAMGIMNS), suggesting a possible role of these peptides in regulating histone modifications in adipose tissue. Several histone markers were identified in a prediabetic mouse model, providing a resource for studying the epigenetic functions of histone modifications in obesity and type 2 diabetes [143]. The expression of a non-specific lipid-transfer protein, also known as sterol carrier protein (2SCP-2), significantly altered the association of several proteins (important for lipid droplet metabolism) with purified lipid droplets both in vitro as well as in intact cells [144]. Therefore, if InPeps from 2SCP-2 that were found to have their levels altered in THOP1^−/−^ could interfere with lipid composition and/or the association of proteins to adipose tissue lipid vesicles, also energy metabolism and fat deposition could be affected. Similarly, InPeps from apolipoprotein A could possess anti-obesity functions [145]. Acyl-CoA-binding protein (ACBP; also known as diazepam-binding inhibitor, DBI) is a lipogenic factor that triggers food intake and obesity [146]. In mice, systemic injection of ACBP protein inhibited autophagy, induced lipogenesis, reduced glycemia, and stimulated both appetite and weight gain. Thus, neutralization of ACBP might constitute a strategy for treating obesity and its co-morbidities [146]. Altered levels of InPeps derived from ACBP have been observed in the adipose tissue of Wistar rats fed a hypercaloric Western diet, and this was shown to improve glucose uptake in 3T3L1 adipocytes [101]. Therefore, InPep substrates and/or products of THOP1 containing post-translational modification or protein-protein interaction motifs, could affect the DIO phenotype of THOP1^−/−^ by differentially regulating protein–protein interactions [60].

Among additional possible mechanisms that InPeps could be affecting energy metabolism, is the modulation of gene expression through binding to specific microRNAs (Table 1) (Figure 6). 

It is known that peptides can bind to pre-microRNAs reducing the levels of mature microRNAs that is formed by DICER [147]. Indeed, in liver and adipose tissues of THOP1^−/−^ animals, either fed SD or HD, alterations were observed in the levels of specific microRNAs, such as miR-127, miR-143, miR-212 and miR-222 [60]. In the liver, higher expression levels of miR-212 and miR-127 were observed in THOP1^−/−^ compared to WT mice fed a standard diet (SD). In liver, an HD increased the expression levels of miR-222 and miR-149 in WT, but not in THOP1^−/−^ mice. In adipose tissue, higher expression levels of miR-143 was observed in THOP1^−/−^ compared to WT mice fed SD. The expression levels of miR-212 and miR-222 increased in WT mice but not THOP1^−/−^ fed HD. In THOP1^−/−^ but not WT mice fed HD, the expression levels of miR-130 were increased, while the expression levels of pri-miR-222 was drastically reduced in the retroperitoneal adipose tissue of THOP1^−/−^ but not in WT mice fed HD [60]. Several InPeps that were differentially quantified in the adipose tissue of THOP1^−/−^ compared to WT fed HD were predicted, by molecular modeling, to interact with both miR-143 and miR-222, but confirmation of this interaction still needs experimental proof. Altogether, these data suggest that THOP1^−/−^ mice have, at least to some degree, altered microRNA expression patterns in their liver and adipose tissues. More importantly, some of these microRNAs were differentially modulated in THOP1^−/−^ fed HD. Genes such as CD36/FAT, fatty acid translocase, also known as cluster of differentiation 36 (CD36), peroxisome proliferator-activated receptor gamma (PPAR-γ), fatty-acid synthase (FAS), lipoprotein lipase (LPL), CD36/SR-B2 (also known as cluster of differentiation 36, platelet glycoprotein 4, or fatty acid translocase/FAT), CD206 (also known as cluster of differentiation 206, C-type lectin mannose receptor), CD11C (also known as cluster of differentiation 11c or integrin alpha X) and murine macrophage F4/80 glycoprotein (F4/80) were among the ones whose mRNA expression were altered in mice lacking THOP1 compared to WT fed HD. Collectively, these data suggest that a dynamic mechanism could be adjusting the fat metabolism of THOP1^−/−^ mice to make them resistant to DIO, insulin resistance and non-alcoholic fatty liver steatosis, when compared to WT mice fed HD (Figure 6). 

The observed changes in additional peptidases, such as dipeptidyl peptidase IV, (DPP4) insulin degrading enzyme, IDE and neprilysin (NEP), could also contribute to the phenotypic differences, as well as modulate InPeps in the adipose tissue of THOP1^−/−^ mice that were fed a HD [60]. The altered expression of specific InPeps in parallel to specific genes and microRNAs corroborates the key role that THOP1 should play in regulating energy metabolism [60].

## 8. THOP1 Relevance in Human Diseases and Diagnostics

THOP1 was associated with Alzheimer‘s disease (AD) as its mapping position on human chromosomes was reported to be within the linkage region for the late-onset AD2 locus on chromosome 19q13.3 [148]. However, after localizing THOP1 to the high-resolution cosmid contig map of human chromosome, results of the hybridization and FISH mapping of positive clones indicated its localization on chromosome 19p13.3. This localization excluded THOP1 from the region that shows evidence of linkage to late-onset familial AD [149]. However, THOP1 was identified in a comprehensive proteomics study aiming at the identification of proteins associated with amyloid-beta (A-beta)-mediated toxicity in cultured cortical neurons. Functional modulation of THOP1 levels in primary cortical neurons demonstrated that its overexpression was neuroprotective against A-beta toxicity, while THOP1 RNA interference (RNAi) knockdown made neurons more vulnerable to amyloid peptide toxicity [150]. In the TgCRND8 transgenic mouse model of amyloid plaque deposition, an age-dependent increase of THOP1 expression was found in brain tissue, where it co-localized with A-beta plaques. In accordance with these findings, THOP1 expression was significantly increased in human AD brain tissue as compared to non-AD controls. These results provide compelling evidence for a neuroprotective role of THOP1 against toxic effects of A-beta in the early stages of AD pathology, and suggest that the observed increase in THOP1 expression might be part of a compensatory defense mechanism of the brain against an increased A-beta load [150]. Lately, THOP1 was further described enriched in protein networks associating it with AD-related biological functions [151]. 

Low THOP1 gene expression was suggested to have clinical potential as a biomarker for better survival prognostic of patients diagnosticated with non-small cell lung cancer (NSCLC) [152]. Quantitative real-time polymerase chain reaction (qPCR) and Western blotting (WB) were employed to quantify the mRNA expression and protein of THOP1 in 16 pairs of primary NSCLC and corresponding normal tissues. Analysis of immunohistochemical staining suggested that low THOP1 expression was found in 71 (59.2%) of the 120 NSCLC specimens and significantly correlated with positive lymph node metastasis (*p* < 0.05). In parallel, low THOP1 expression was found only in 22 (41.5%) of the 53 normal lung tissues. THOP1 expression was significantly higher in the normal lung tissues than that in the NSCLC specimens (*p* < 0.05). Quantitative real time PCR (qRT-PCR) and WB showed that NSCLC specimens had decreased THOP1 mRNA and protein expression compared to corresponding normal tissues. Low THOP1 expression significantly predicted a decreased 5-year disease-free survival (*p* < 0.05) and overall survival (*p* < 0.02). In addition, low THOP1 expression significantly predicted decreased 5-year overall survival in lymph node metastasis (*p* = 0.025) and advanced TNM stage (*p* = 0.009; TNM meaning: T, extent of the tumor, N, the extent of spread to the lymph nodes, and M, the presence of metastasis) [152]. Conversely, a different study that corroborates these findings showed that high expression of THOP1, together with additional proteins, was associated with poor prognosis of lung adenocarcinoma [153].

Lower gene expression of THOP1 in the background liver of hepatocellular carcinoma (HCC) was also suggested to be a biomarker for risk of HCC development [154]. Real-time reverse transcriptase PCR (qRT-PCR) was used to evaluate the expression of THOP1 in different human liver samples. Control samples, termed supernormal (SN) liver, were taken from 11 cases of metastatic secondary malignancies of the liver. Adjacent non-neoplastic liver tissue from a patient with HCC and liver cirrhosis by hepatitis C (CN) were also evaluated for comparison. Expression profiling showed that expression of THOP1 gene was decreased four-fold in CN. THOP1 gene expression by qRT-PCR decreased in matching normal tissue relative to SN. The group with higher than average THOP1 expression (*n* = 74) showed significant correlations with prolonged survival (*p* = 0.0383). Strongly reduced THOP1 expression (<3.0, *n* = 50) was shown to be an independent prognostic factor by multivariate analysis (*p* = 0.0024).

Previous genetic association studies integrated with gene expression and pathway analyses identified THOP1 as a relevant protein for the development of rheumatoid arthritis (RA) [155]. RNA-sequencing-based expression analysis of 377 genes from previously verified RA-associated loci was performed in blood cells from 5 newly diagnosed, non-treated patients with RA, 7 patients with treated RA and 12 healthy controls. Differentially expressed genes sharing a similar expression patterns in treated and untreated RA sub-groups were selected for pathway analysis. A set of “connector” genes derived from pathway analyses was tested for differential expression in the initial discovery cohort and validated in blood cells from 73 patients with RA and in 35 healthy controls. THOP1 whole blood RNA expression was significantly lower in both treated and non-treated patients with RA, compared to healthy controls. THOP1 gene expression was also negatively correlated to DAS28-C-reactive protein (CRP) score, in peripheral blood mononuclear cells from 43 untreated female patients with RA [155]. THOP1 expression profile was successfully replicated in RNA-seq data from peripheral blood mononuclear cells from healthy controls and non-treated patients with RA, in an independent collection of samples. Integration of RNA-seq data with findings from association studies, and consequent pathway analyses implicate THOP1 as a new candidate gene in the pathogenesis of RA [155]. The downregulation of THOP1 in whole blood and peripheral blood mononuclear cells from patients with RA could result in abnormal antigen presentation, which might contribute to the pathogenesis of RA and inflammatory diseases [155]. 

Indeed, THOP1^−/−^ mice showed increased percentages of granulocytes and monocytes and a decreased percentage of lymphocytes in blood, which could also contribute to the differences seen in the autoimmune response. However, additional physiological functions contributed by THOP1^−/−^ mice also included a better survival and performance behavior seven days after polymicrobial sepsis induction, with a slightly lower expression of TLR4 and TNF-α in dorsal hippocampus following the sepsis [61]. Sepsis is an organ dysfunction caused by a deregulated host response to infection which is often accompanied by an intense systemic inflammatory response [156,157]. These data suggest that THOP1^−/−^ mice could be an interesting model for future investigations on the role of THOP1 in the development of cancer and inflammatory diseases.

## 9. Chemical Inhibitors of THOP1

The development of stable and selective inhibitors of THOP1 have been tried along the years. One of the first THOP1 inhibitors that has been developed, CFP, was the most widely studied [158]. CFP is a potent and specific inhibitor, but it is unstable in vivo due to cleavage between the alanine and tyrosine residues by NEP [159]. This cleavage by NEP generates a potent inhibitor of angiotensin converting enzyme 1 (ACE1), thereby limiting the use of CFP for in vivo studies. Additional inhibitors of THOP1 based on the CFP were developed such as *N*-[1-(*R*, *S*)-carboxy-3-phenylpropyl]-A-Aib-Y-p-aminobenzoate (JA2), which was resistant to in vivo proteolysis after incorporation of beta-amino acids [117,127,160]. JA2 is a mixed inhibitor of THOP1 and the structurally homologous enzyme neurolysin (EC3.4.24.16; Nln), which was resistant to cleavage by the related metalloendopeptidase NEP, in contrast to the precursor CFP inhibitor [117,127,160]. The potent pharmacological in vivo effects of JA2 was seen in the hypotension induced by bradykinin, which did not diminish even 4 h after JA2 injection [127]. In contrast, the hypertensive effects of angiotensin I and II were unaltered by JA2, indicating that the bradykinin-potentiating effects were not due to ACE1 inhibition. These data suggest that JA2 was not only a potent and specific inhibitor of THOP1/Nln but also was stable in vivo [127]. 

Numerous phosphinic peptides were also synthesized and shown to act as potent (Ki in the nanomolar range) THOP1 inhibitors [161]. THOP1 was observed exhibiting a marked preference for phosphinic inhibitors containing a basic residue (R or K) in the P2’ position (corroborating the partial sub-site preference of THOP1 for substrates containing Lys at P2’; Figure 2). The compound *Z*-(l,d)Phe psi (PO_2_CH_2_)(l,d)ARM (mixture of the four diastereoisomers) displays a Ki value of 70 pM for THOP1, but was also a mixed inhibitor of Nln. The most potent (Ki value of 0.160 nM) and selective phosphinic inhibitor was the compound *Z*-(l,d)Phe psi (PO_2_-CH_2_)(l,d)ARF. This latter phosphinic inhibitor was more than 3 orders of magnitude less potent toward Nln (Ki = 530 nM), and concentrations up to 1 µM were unable to affect the activity of several other zinc peptidases [161]. In vitro, *Z*-(l,d)Phe psi (PO_2_CH_2_)(l,d)ARF is still considered the most potent and specific inhibitor of THOP1 to date [161].

Recently, an unusual alosteric mixed inhibitor of THOP1 and Nln that can be used to inhibit these enzymes in cells and in vivo was reported [162]. The inhibitor compound *3-[(2S)-1-[(3R)-3-(2-Chlorophenyl)-2-(2-fluorophenyl)pyrazolidin-1-yl]-1-oxopropan-2-yl]-1-(adamantan-2-yl)urea* (R2), showed relatively potent inhibitory activity against THOP1 and Nln (*p*IC50 value of 7.5 ± 0.14). The R2 inhibitor was shown by crystallography to bind remotely from the catalytic site of Nln, which suggested that its inhibition must occur through an allosteric mechanism [162]. Recently, R2 was used in cells and in vivo showing chemical inhibition of THOP1/Nln, which impaired oxidative metabolism and disrupted the formation of respiratory chain supercomplexes [163]. Primary acute myeloid leukemia (AML) and normal hematopoietic cells were injected into the femurs of immunodeficient mice. Two weeks later, these mice were treated with R2 or vehicle control. Treatments with R2 reduced the leukemic burden in these mice without toxicity [163]. Therefore, R2 targeted the AML stem cells, as evidenced by decreased engraftment in secondary experiments. In contrast, R2 did not reduce the engraftment of normal hematopoietic cells. These results demonstrate that pharmacological inhibition of THOP1/Nln with the allosteric inhibitor R2 impairs leukemic cell growth in vitro and in vivo [163]. Future studies should allow the development of novel and more specific inhibitors of THOP1, which could be tested for clinical applications.

## 10. Concluding Remarks and Future Perspectives

Because THOP1 is mainly located within cells, proteasome activity is suggested to produce most of its intracellular substrates. As previously mentioned, THOP1 has strict substrate-size specificity, not degrading proteins, suggesting that its biological functions should be mediated by peptides. Therefore, the major biological significance of THOP1 seems to occur by the following mechanism: THOP1 acting on peptides that were previously generated by proteasomal activity, controls specific InPep profiles in different tissues. Specific InPeps generated by the proteasome and further processed by THOP1 (or other intracellular peptidases such as Nln), can regulate protein-protein interactions and microRNA processing. These functions of InPeps could be affecting gene expression, and consequently, protein levels. Once protein levels are altered, their macromolecular interactions are subsequently modified, affecting the phenotype [164,165,166,167] (Figure 7). It seems relevant to mention that synthetic peptides identified by phage display have been previously shown to affect DICER pre-microRNA processing, reducing the levels of the mature microRNAs formed [147]. The ubiquitous existence of proteasome, THOP1, InPeps and microRNAs in different cells of different species, suggests a broader biological significance for this mechanism. 

A previously non anticipated link between proteasomal-protein degradation and protein expression is suggested by this mechanism, which could be of great relevance for most diseases involving alterations of proteasomal and/or THOP1 activity function. In the adipose tissue of mice fed SD, most of the relative ratios of InPeps in THOP1^−/−^/WT were increased; when these mice were fed HD, most of the relative ratios of InPeps in THOP1^−/−^/WT were decreased [60]. As demonstrated, HD affects the expression of other major intracellular peptidases such as prolyl oligopeptidase and insulin degrading enzyme in THOP1^−/−^. InPeps that were increased in SD-fed THOP1^−/−^, but decrease in THOP1^−/−^ fed HD, could be substrates of prolyl oligopeptidase and/or insulin degrading enzyme. These suggestions were not yet investigated but could implicate that alterations in the levels of additional intracellular peptidases could affect InPep profile and function.

Alzheimer‘s disease, non-small cell lung cancer and hepatocellular carcinoma were all described influencing THOP1 expression, which may be an indicative that THOP1 and InPeps are related to human pathologies. In fact, previous results have already indicated alterations of specific InPeps in both the anterior temporal lobe (ATL) and corpus callosum (CC) of schizophrenic patients [87]. The levels of one of these InPeps (SEGTKAVTKYTSSK, PepH), a fragment of histone H2B type 1-H, is significantly decreased in the ATL of schizophrenic patients. H2B type 1-H histone, along with other histones (HIST1H2BD, HIST1H2BC, HIST1H2BG, and HIST1H4K), is localized in a region of the MHC known as the extended MHC (xMHC). The xMHC is a region of approximately 8 Mb on chromosome 6 that contains around 250 codifying genes. According to genome-wide association studies (GWAS), the xMHC chromosome has one of the major correlations with schizophrenia [168]. PepH was tested in serum-deprived Neuro2A cells and showed a protective effect against cell death. Also, in Neuro2A cells that were challenged with lipopolysaccharide (LPS), PepH was able to prevent the endotoxic effects of LPS. These data suggest that specific InPeps are altered in schizophrenia patients, and that InPeps could be novel targets in studies not only of schizophrenia but also of other neuropsychiatric diseases.

Exciting future investigations will likely demonstrate the feasibility of the mechanisms proposed herein, which may suggest that InPeps mediate at least part of protein functions following proteasomal degradation.

## Figures and Tables

**Figure 1 biomolecules-10-01229-f001:**
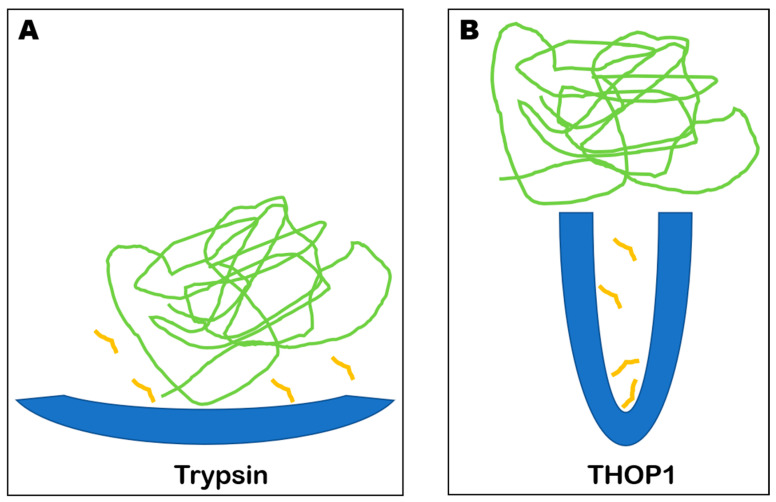
Schematic representation of trypsin and THOP1 catalytic cleft. (**A**) trypsin catalytic cleft is located on the enzyme surface, which allows some similar proteases to cleave large structured or unstructured proteins as well as relatively small peptides. (**B**) THOP1 catalytic cleft, distinctively from proteases such as trypsin, is located at the bottom of a deep and narrow channel. This structural organization makes proteins resistant to THOP1 cleavage. Oligopeptidases such as neurolysin (Nln) and prolyl-oligopeptidase (POP) also have similar substrate-size restrictions as THOP1. As far as it is known, only specific peptides in the size range of 5–22 amino acids have been described as substrates of THOP1. Blue figures, illustrates the enzymes trypsin and THOP1; small yellow lines, illustrates peptides; long green lines, illustrates a folded protein.

**Figure 2 biomolecules-10-01229-f002:**
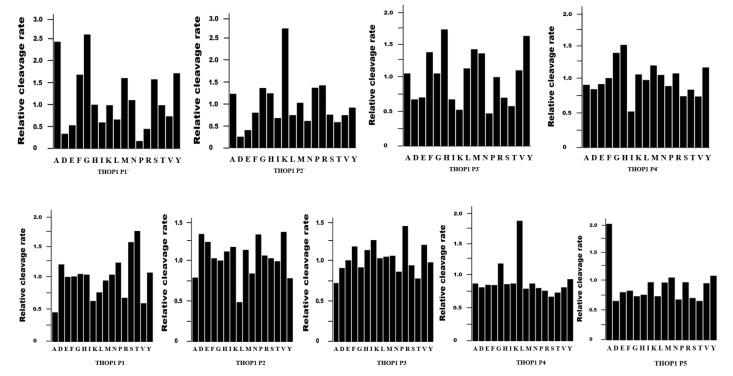
Analysis of the specificity of recombinant THOP1 regarding the enzyme preference for certain amino acids as potential substrates, positioned in specific sub-sites (P’ or P, defined according to Schechter and Berger) [39], using peptide libraries. Experiments were carried as previously described [41]. Briefly, for enzyme–peptide combinations in which the reaction rate was linear over substrate concentration [S] at 100 μM, values of *k*_cat_/*K*_M_ were determined from initial rates (<10% turnover) at that concentration (where *K*_M_ ≫ [S]). Catalytic parameters were obtained by determining initial rates at various substrate concentrations and fitting the data directly to the Michaelis–Menten equation using Kaleidagraph software [41]. Assays were performed in triplicate that varied less than 10% among each other. Enzyme concentrations used were based on protein concentration alone of homogeneously purified rat testis recombinant THOP1 [43].

**Figure 3 biomolecules-10-01229-f003:**
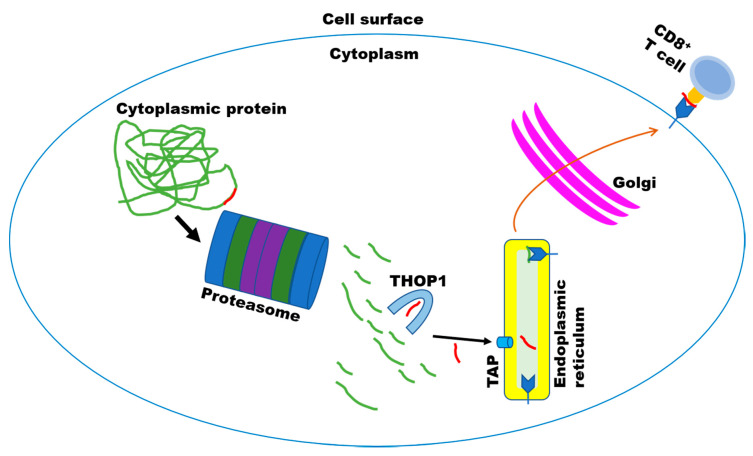
The major histocompatibility complex of class I (MHC-I) processing pathway. Most frequently only one peptide *per* protein is used as antigen presented by MHC-I (represented herein as the red line “peptide”). Note, this is the same proteasomal activity route that generates hundreds of additional non-antigenic intracellular peptides (InPeps; represented herein as green lines “peptides”). Self and viral proteins are degraded by the proteasome into peptides, which can be further processed by THOP1 to either inhibit or enlarge MHC-I antigen processing depending on the antigen [58]. A small fraction of these proteasome and THOP1 processed InPeps are transported into the endoplasmic reticulum lumen by transporter associated protein (TAP). Upon binding of only one of these TAP-transported peptides to MHC-I heterodimers in the lumen of the endoplasmic reticulum, the complex is transported to the plasma membrane where it is presented to the immune system through CD8^+^ T cells.

**Figure 4 biomolecules-10-01229-f004:**
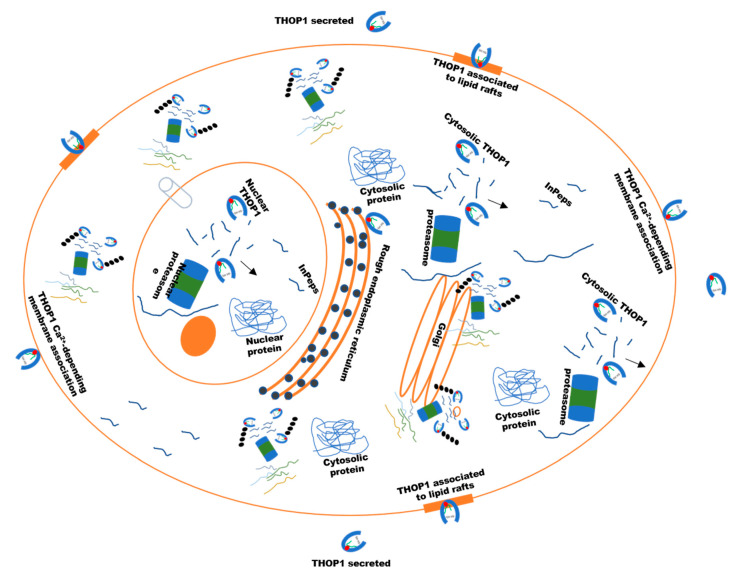
THOP1 cell distribution has implications for its function on peptide metabolism. THOP1 and proteasome shares similar subcellular distribution within cells, present free in the cytosol, associated with the cytoplasmic face of the endoplasmic reticulum and inside the nucleus [49,122,123]. These suggests that proteasome and THOP1 in concert can function degrading, respectively, proteins and peptides within cells. THOP1 has also an additional extracellular function as a secreted and membrane-associated enzyme, metabolizing neuropeptides such as opioids, bradykinin, neurotensin and gonadotrophin-releasing hormone (GnRH). THOP1 can be found in lipid rafts and associated to the plasma membrane in a Ca^2+^-dependent manner.

**Figure 5 biomolecules-10-01229-f005:**
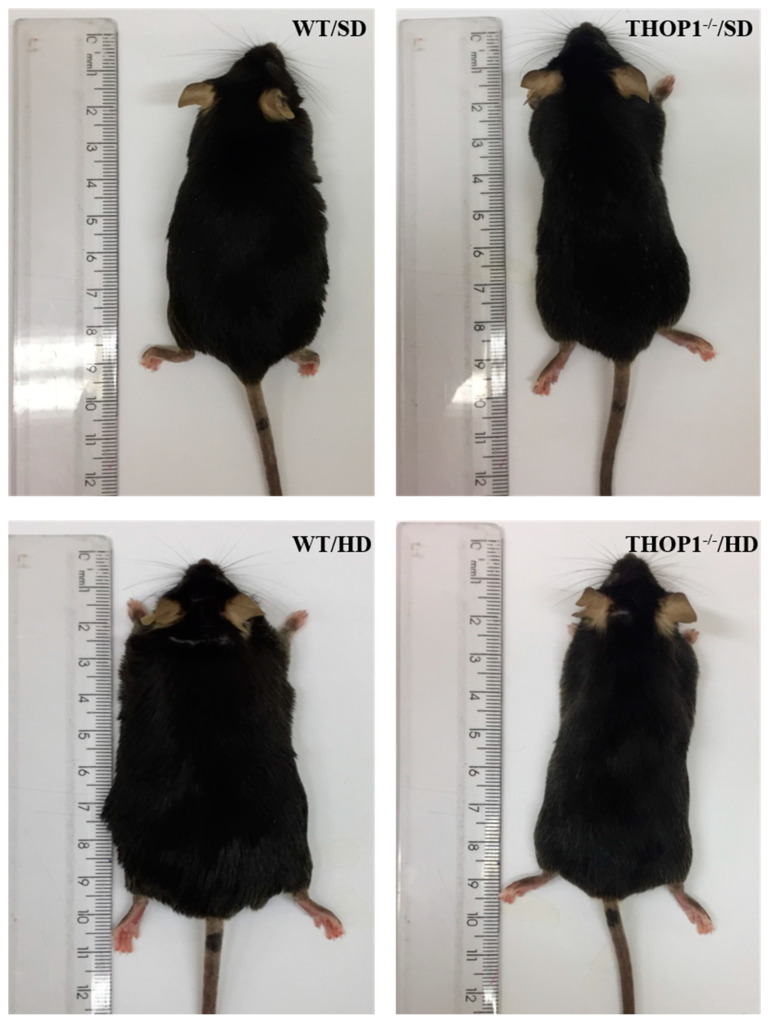
Visual appearance of wild-type (WT) and THOP1^−/−^ animals fed either a standard (SD) or a hyperlipidic (HD) diet after 24 weeks. The high fat content of WT mice compared to THOP1^−/−^, both fed HD, was confirmed by X-ray density images and adipose tissues weighting [60]. Male WT animals fed a HD gained an extra 14.2 g of body weight when compared to WT males fed an SD. Male THOP1^−/−^ animals fed a HD during the same period gained only 3.8 g compared to THOP1^−/−^ males fed an SD, corresponding to only 27% of the body weight gained by WT animals fed a HD [60]. THOP1^−/−^ mice were healthy and even have a better performance on the high-intensity treadmill test compared to WT mice; suggesting the improved oxidative metabolism of THOP1^−/−^ mice.

**Figure 6 biomolecules-10-01229-f006:**
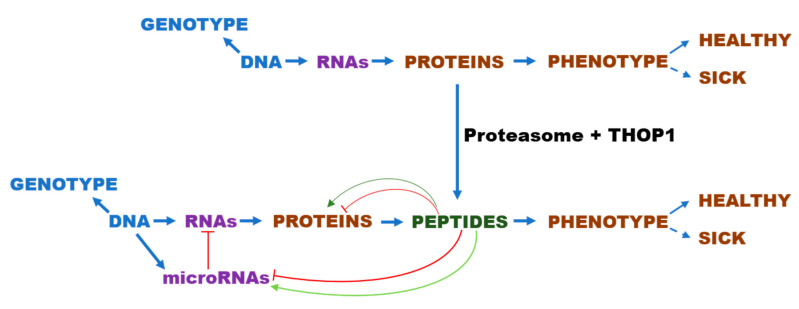
Knocking out THOP1 expression was a genotype alteration that alone largely ameliorated C57BL6/N mice diet-induced obesity (DIO) phenotype. An exciting novel mechanism for controlling body weight was suggested, where THOP1 can act after the proteasome to control the intracellular peptides (InPeps) profile. InPeps generated by proteasomes and further processed by THOP1 can control protein interactions and the levels of microRNAs by a yet under investigation mechanism. Consequently, differential expression of specific microRNAs and altered protein-protein networks, can control the expression of genes to improve the oxidative metabolism of THOP1^−/−^ mice.

**Figure 7 biomolecules-10-01229-f007:**
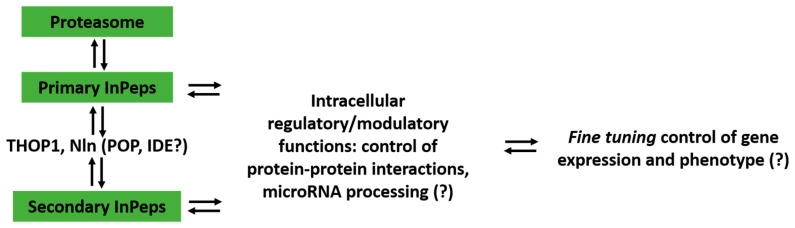
THOP1^−/−^ mice have a genotype alteration that is not lethal but causes some phenotype alterations, including a great resistance to hyperlipidic diet-induced obesity (DIO). A general novel mechanism for controlling healthy and unhealthy phenotypes could be taking place in these THOP1^−/−^ animals, where the lack of THOP1 acting after the proteasome control specific InPep profiles. InPeps regulate protein interactions and the expression of specific microRNAs, consequently, shifting gene expression and affecting macromolecular organization of proteins networks. Neurolysin, Nln; prolyl oligopeptidase, POP; insulin degrading enzyme, IDE. (?), under investigation.

**Table 1 biomolecules-10-01229-t001:** mRNA and microRNAs differentially expressed on THOP1^−/−^ [60,61].

mRNA	microRNA	THOP1^−/−^ Effect on Expression
ACE1	nd	higher (ST and HC)
NEP	nd	higher (ST); lower (female retroperitoneal adipose tissue)
Proteasome beta5 subunit	nd	lower (ST and PFC); lower (female retroperitoneal adipose tissue)
DPP4	nd	lower (female retroperitoneal adipose tissue)
POP	nd	lower (female retroperitoneal adipose tissue)
IDE	nd	higher (male retroperitoneal adipose tissue)
5HT receptor	nd	higher (HC)
Dopamine DR2D receptor	nd	lower (ST and HC)
Adrenergic beta 1 receptor	nd	higher (male retroperitoneal adipose tissue)
Adrenergic beta 3 receptor	nd	higher (male retroperitoneal adipose tissue)
Adrenergic beta 3 receptor	nd	lower (female retroperitoneal adipose tissue)
CD36	nd	lower (male liver tissue; male/female inguinal adipose tissue)
LPL	nd	lower (male/female inguinal adipose tissue)
CD206	nd	lower (male/female inguinal adipose tissue)
CD11C	nd	lower (male inguinal adipose tissue)
CD11C	nd	higher (female inguinal adipose tissue)
F4/80	nd	lower (male/female inguinal adipose tissue)
PPAR-γ	nd	higher (male inguinal adipose tissue of THOP1^−/−^ fed HD)
FAS	nd	higher (male inguinal adipose tissue of THOP1^−/−^ fed HD)
	miR-212	higher (male liver tissue)
	miR-127	higher (male liver tissue)
	miR-143	higher (male retroperitoneal adipose tissue)
	miR-222	unaltered THOP1^−/−^ fed HD (male retroperitoneal adipose tissue)

**Footnote:** not determined, nd; microRNA, miR; angiotensin converting enzyme 1, ACE1; neprilysin, NEP; dipeptidyl peptidase IV, DPP4; prolyl oligopeptidase, POP; insulin degrading enzyme, IDE. CD36/FAT, fatty acid translocase, also known as cluster of differentiation 36 (CD36), lipoprotein lipase (LPL), cluster of differentiation 206, C-type lectin mannose receptor (CD206), cluster of differentiation 11c or integrin alpha X (CD11C), murine macrophage F4/80 glycoprotein (F4/80), peroxisome proliferator-activated receptor gamma (PPAR-γ), fatty-acid synthase (FAS).

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
