# Peer review of "Thimet Oligopeptidase Biochemical and Biological Significances: Past, Present, and Future Directions"

_biomolecules, 2020, doi:10.3390/biom10091229_

Round 1

Reviewer 1 Report

The review "Thimet oligopeptidase biochemical and biological significances: past, present and future directions." by Ferro and is focused on the discussion of data concerning the thimet oligopeptidase THOP1, which is involved in major histocompatibility class I antigen presentation, hyperlipidic diet-induced obesity, insulin resistance and non-alcoholic liver steatosis.

In principal,  the topic of the  review is of general interest and the manuscript presents many important informations. However, there are three major pounts that have to be olved by the authors.

1) The manuscript text is not always well written. Examples include the abstract ("Altogether, investigation of THOP1 biochemical and biomedical significances let to the speculation that degradation by proteasome may not be the end of protein function. But rather a processing step, which allow
functional proteasome activity products to control RNA translation and protein function.") or several text passages like line 301 ("However, another exciting possibility also exists and is as follows."). Therefore, extensive editing of English language and style are required.

2) The figures are not well referenced and described in the text. The table 1 still has the Word Template line "Table 1. should be around here." (line 412).

3) Figure 2: The authors include unpublished original data in this review. However, if the want to do so, they should present them in statistically correct way. This means that they should give the information how many times the measurements were repeated and they should include the standard eror of mean / error bars. 

The authors have to solve the mentioned points.

Author Response

Dear reviewer,

We would like to thank you immensely for the time and attention you spent in reviewing our article. Below, find the point-to-point answers to your questions. We hope to have answered them properly. However, do not hesitate to send new suggestions and criticisms if necessary. We conducted a review of English with a professional native speaker.
Sincerely,

Emer S. Ferro, 

corresponding author

Reviewers major questions and, ours point-to-point answers:

"However, there are three major points that have to be solved by the authors."

1) The manuscript text is not always well written. Examples include the abstract ("Altogether, investigation of THOP1 biochemical and biomedical significances let to the speculation that degradation by proteasome may not be the end of protein function. But rather a processing step, which allow functional proteasome activity products to control RNA translation and protein function.") or several text passages like line 301 ("However, another exciting possibility also exists and is as follows."). Therefore, extensive editing of English language and style are required.

Response: We thank the reviewer for mentioning these weaknesses of the original version of our manuscript. We went through an extensive revision of the English both by ourselves and by a professional English editor. We hope the revised version of the manuscript can reaches the expectation of the reviewer.

2) The figures are not well referenced and described in the text. The table 1 still has the Word Template line "Table 1. should be around here." (line 412).

Response: We have made the appropriate changes on the original version to fix the points mentioned by the reviewer.

3) Figure 2: The authors include unpublished original data in this review. However, if the want to do so, they should present them in statistically correct way. This means that they should give the information how many times the measurements were repeated and they should include the standard eror of mean / error bars.

Response: We thank the reviewer for bringing these comments. We have now included additional information on the figure legend, which includes mentioning the smaller than 10% variation on the average of the kinetic results obtained. We have also included more details about the methodology that those results were obtained. We hope o have provided the additional information needed to maintain these original data as part of this manuscript.  

Additionally, we have also made general changes in the manuscript trying to address the general comments from the reviewers regarding the weak points on “work significant contribution to the field”, “work organization and comprehensively” and “work scientifically sound and not misleading”. 

Reviewer 2 Report

The review manuscript by E.S. Ferro and A. Navon entitled „Thinmet oligopeptidase biochemical and biological significances; past, present and future directions” provides an interesting overview on THOP1, a metallopeptidase whose function is linked with several vital processes in the organism. Overall, the text is well written and includes the majority of recent publications on THOP1, as well as historical ones.

There are some issues that should be addressed by the Authors to improve the manuscript:

1. Figure 1 A – I do not agree that surface localization of active centers is characteristic for most of proteases. This feature may be attributed for example to digestive proteases (eg. trypsin) or „destructive” proteases ( eg. caspases) but there are numerous proteases whose active centers are not easily accessed (eg. self-compartmentalizing proteases – proteasome or several bacterial proteases that form cage-like oligomers). I find the drawing of trypsin is a bit exaggerated – the active center is place in the cleft on the protein surface and the active triad does not protrude outside the molecule.

What does the „red dot” in the picture 1A show?

P and P' were not shown in the figure.

2. The text should be supplemented with illustrations depicting the processes in which THOP1 is involved (including the proposed role of THOP1), ideally in each chapter. This would make the text much easier to follow.

Author Response

Dear reviewer,

We would like to thank you immensely for the time and attention you spent in reviewing our article. Below, find the point-to-point answers to your questions. We hope to have answered them properly. However, do not hesitate to send new suggestions and criticisms if necessary. We conducted a review of English with a professional native speaker.
Sincerely,

Emer S. Ferro, 

Reviewers major questions and, our point-to-point answers:

Comments and Suggestions for Authors

The review manuscript by E.S. Ferro and A. Navon entitled „Thinmet oligopeptidase biochemical and biological significances; past, present and future directions” provides an interesting overview on THOP1, a metallopeptidase whose function is linked with several vital processes in the organism. Overall, the text is well written and includes the majority of recent publications on THOP1, as well as historical ones.

 There are some issues that should be addressed by the Authors to improve the manuscript:

  1. Figure 1 A – I do not agree that surface localization of active centers is characteristic for most of proteases. This feature may be attributed for example to digestive proteases (eg. trypsin) or „destructive” proteases ( eg. caspases) but there are numerous proteases whose active centers are not easily accessed (eg. self-compartmentalizing proteases – proteasome or several bacterial proteases that form cage-like oligomers). I find the drawing of trypsin is a bit exaggerated – the active center is place in the cleft on the protein surface and the active triad does not protrude outside the molecule.

What does the „red dot” in the picture 1A show?

P and P' were not shown in the figure.

Response: We thank the reviewer for mentioning these weaknesses of the original version of Figure 1. We completely agree with the review and want to apologize for not making it clear on the legend of the original figure. We revised the figure and the legend to emphasize that the superficial catalytic site meant a characteristic of trypsin. The red dot was originally meant to indicate the catalytic site, but it was not essential and was removed from the actual figure, without compromising the information that we intend to give the readers. We have added information about the origin of the P and P` definition for the subsites of interaction between enzyme and substrates.

  1. The text should be supplemented with illustrations depicting the processes in which THOP1 is involved (including the proposed role of THOP1), ideally in each chapter. This would make the text much easier to follow.

Response: Again, we want to thank the reviewer for mentioning these weaknesses of the original version of our manuscript. We added several additional figures hoping to reach the expectations of the reviewer.

Additionally, we have also made general changes in the manuscript trying to address the general comments from the reviewers regarding the weak points on “work significant contribution to the field”, “work organization and comprehensively” and “work scientifically sound and not misleading”. 

Round 2

Reviewer 1 Report

The authors have addressed the points in the revised version. The manuscript can be accepted for publication.